# The impact of shoreline change on the salinity distribution in the wetlands of Liao River Estuary, China

Mingliang Zhang[1], Tianping Xu[1, 2], Hengzhi Jiang[3]

[1]School of Marine Science and Environment, Dalian Ocean University, Dalian, China;

[2]School of Marine Sciences, Sun Yat-sen University, Guangzhou, China

[3]National Marine Environment Monitoring Center, Dalian, China

Corresponding to: Hengzhi Jiang (jianghengzhi99@163.com)

**Abstract:** The wetland of Liao River Estuary in northeast China is one of the best-preserved wetlands across the globe. However, it is extremely vulnerable to hydrological changes as well as other disturbances, particularly upstream river discharges and the reclamation from anthropogenic activities. In this study, a 3D hydrodynamic model (FVCOM) was used to reproduce the flow patterns of the Liao River Estuary and to explore the variation in salinity under different scenarios. Furthermore, the impact of river discharge and shoreline changes on the salinity distribution in the Liao

River Estuary were quantitatively analyzed and discussed through several simulation experiments. The model reasonably reconstructed the spatio-temporal variability and distribution of salinity in the Liao River Estuary and the wetlands across intertidal areas. Increases in runoff were demonstrated to significantly decrease the mean salinity values of the estuary, with changes in salinity negatively correlated to the longitudinal distance from the estuary mouth. Moreover, the shoreline change caused by the construction of Panjin Port was observed to have an obvious influence

on the salinity distribution of the Liao River Estuary, particularly in the lower reaches of the Pink Beach Wetland. Comparisons of the Liao River Estuary residual flow fields under different shorelines revealed that the establishment of the port resulted in the diffusion of the runoff from the Daliao River due to the obstruction of the port body, which enhanced the tidal mixing effect and consequently weakened the dilution effect of fresh water entering the Pink Beach Wetland. Overall, the current study should be helpful for offering a greater understanding of *Suaeda heteroptera*
vegetation degradation in the Liao River Estuary, China, this work also provides a new perspective in investigating the

degradation mechanism in other estuarine wetlands.

## 1 Introduction

Estuaries are located at the interface area of rivers, land and the ocean, and are commonly described as enclosed or

semi-enclosed bodies of water, where sea water is significantly diluted by the inflow of freshwater from upstream

rivers. Accumulated freshwater can potentially regulate water and salinity at different temporal scales, and can offer

the possibility to optimize suitable salinity conditions for animal and plant habitats within the estuarine zone. Estuaries

are thus considered as highly productive ecosystems. As an essential component of estuaries, estuarine wetlands

function as the bond connecting terrestrial and coastal water ecosystems, possessing features from both terrestrial and

aquatic environments. Estuarine wetlands can provide substantial ecological services, including flood control,

sediment retention, the uptake of pollutants, carbon fixation and also act as habitats for fish, waterfowl and other biotic

communities (Tsihrintzis et al., 1998; Weilhoefer, 2011; Yang et al., 2018). Hence, tidal wetlands hold the unique role

of maintaining the ecological equilibrium and protecting biodiversity (McClain et al., 2003).

Estuaries form an environment with sinuous coastlines, where rivers, streams, or creek inlets meet the sea or ocean.

The hydrological properties of estuaries create a complex physical environment due to the combined effects of

upstream rivers and the adjacent ocean. Estuary circulations are primarily controlled by freshwater discharge, the tidal

range and tidal currents (Sassi and Hoitink, 2013; Veerapaga et al., 2019), and are modified by winds, shorelines and

topography (Alebregtse and de Swart, 2016; Lai et al., 2018), which further influence the processes of salinity

transport and salt-wedge in estuary waters (Haralambidou et al., 2010; Gong and Shen, 2011; Liu et al., 2019). Salinity

changes in estuaries have been the focal point of much work over the past few decades in order understand saltwater

intrusion mechanisms, with the majority of the research based on on-site observations and numerical models. Early

studies analyzed observation data to investigate the salt intrusion phenomenon in estuaries, with further evaluations

performed on estuary mixing type, salt intrusion distance, salinity distribution and the current circulation (Pichard

1952, 1954; Hansen and Rattray, 1965). This was followed by field surveys via instrumentation of Acoustic-Doppler-Current-Profilers (ADCP) and Conductivity-Temperature-Depth (CTD), whereby researchers investigated the salinity distribution characteristics and the salt transport mechanisms in the Hudson River estuary (Bowen and Geyer, 2003; Lerczak and Geyer, 2006). Studies suggested that the salt intrusion in an estuary is maintained by two opposite

longitudinal transport processes: i) An advection resulting from fresh water outflow, which tends to drive salt out of the estuary; and ii) a down gradient advection, which drives salt landward (Lerczak and Geyer, 2006). The salinity patterns in estuaries were also demonstrated to be the result of additional factors, such as the longitudinal advection and vertical mixing (MacCready, 2010; Scully and Geyer, 2012).

The development and application of numerical models have been a key component of research on the salt transport and

saltwater intrusion in estuarine waters at high spatial and temporal resolutions. For example, Zhao et al. (2010) simulated wetland-estuarine-shelf interaction processes at the Plum Island Sound and Merrimack River system in the Massachusetts coast using the three-dimensional (3D) unstructured-grid, finite volume coastal ocean model (FVCOM), where the water exchange was observed to vary with the magnitude of freshwater discharge and wind conditions. Gong (2011, 2018) employed ELCIRC and COAWST to investigate the effects of local and remote winds and wind

waves on salt intrusion in the Modaomen and Pearl River estuaries, respectively, with conclusions having an applicability to other partially mixed estuaries under the threat of salt intrusion. Popescu et al. (2015) presented a 3D hydrodynamic model to understand the hydrodynamics of the Danube Delta wetlands, demonstrating the strong impact of the Danube River variable discharges on the habitats and the overall ecological status of the delta. Andrews et al (2017) applied the 3D hydrodynamic model UnTRIM to examine the impacts of anthropogenic changes (land use

changes, levee construction, channel modifications etc.) on the physical characteristics and salt intrusion processes at the San Francisco Estuary. Wang et al (2019) combined the FVCOM model with long-term field survey data to investigate the seasonal variability of currents and salinity in the Indus River Estuary. Results revealed that the salinity intrusion distance was principally controlled by river discharge in the estuary, while the seawater intrusion was significantly impacted by tidal forcing in the absence of river runoff. Tian (2019) combined field data collected in the

Chester River Estuary with FVCOM to identify the driving factors and their relative importance in controlling the saltwater intrusion variability across time. Results demonstrated that river discharge was the primary factor controlling saltwater intrusion changes over inter-annual time scales, while sea surface levels dominated the seasonal variations (Tian, 2019).

Estuarine salinity has a significant effect on the growth of coastal wetland plants and plays an important role in maintaining the ecological health of estuarine wetlands. Despite this, studies on the spatial and temporal distribution of salinity in estuarine wetlands are limited, with most work generally focusing on salinity transport and saltwater intrusion mechanisms along the estuaries.

The Liao River (LR) in northeastern China is one of the seven largest rivers in the country and ends at the Liao River

Estuary (LRE) located at the top of Liaodong Bay (LDB). *Suaeda heteroptera* (*S.h.*) and *Phragmites australis* (*P.a.*) are the most common pioneer salt-tolerant plants in the LRE wetlands. According to satellite images and on-site surveys, the wetlands in the LRE have experienced severe degradation over the past decade. Much effort has been made to investigate the growth mechanisms and influencing factors of *S.h.* degradation in salt marshes (Song et al., 2009; Wang et al., 2010; Sun et al., 2016). Studies have revealed that salinity increases in water and soil can result in

the death of *S.h.* vegetation.

Panjin City, close to LR and the Daliao River (DLR) is listed as the key development node and import and export port of grain logistics in the Circum-Bohai-Sea area. Marine land reclamation along the coast of LRE initiated in 1995 and is on-going in order to build the Panjin sea port. Landsat images of the LRE region collected in 1995 and 2019 (Fig. 1) show the immense changes of the southeast coastline in the LRE due to the construction of the port. The Landsat data

was downloaded from Geospatial Data Cloud Site, Computer Network Information Center, Chinese Academy of Sciences (http://www.gscloud.cn/sources/?cdataid=263&pdataid=10). Soil and water salinity are crucial for vegetation growth in the wetlands of LRE. However, the impacts of shoreline changes on the temporal and spatial distributions of salinity in the LRE, and the relationship between wetland degradation and port construction, are not yet fully understood.

In this study, the FVCOM system was combined with field observations to investigate the effect of anthropogenic shoreline changes on the salinity distribution in the LRE. The principle aims of this study are to: (1) Evaluate and quantify the effect of shoreline changes resulting from the port construction on estuary circulations and salinity distributions in the LRE; (2) explore the internal mechanisms of these effects; and (3) provide a reference for wetland

conservation and water resource utilization in the LRE. The rest of the paper is organized as follows. A brief introduction of the study area is described in Section 2. Section 3 provides a description of the FVCOM model and demonstrates the implementation of the modeling system and model validation, including tidal level, flow and salinity. The effects of shoreline change resulting from port construction on residual flow and the spatial-temporal distribution of salinity in the LRE are analyzed and discussed in detail in Section 4. Finally, the general conclusion of the

manuscript is detailed in Section 5.

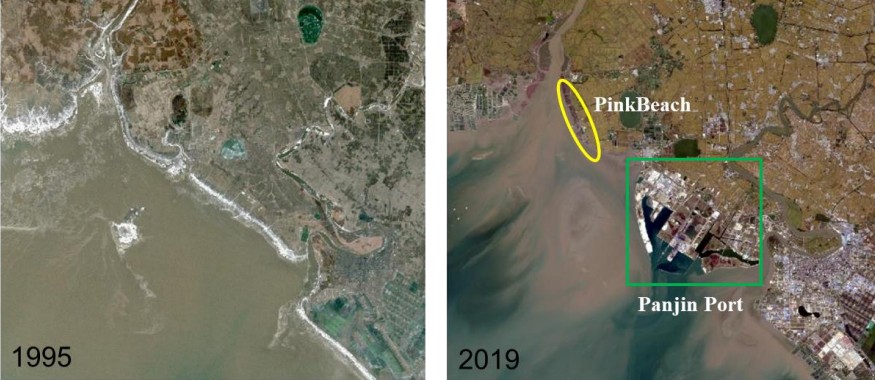

**Figure 1** Shorelines of the Liao River Estuary in 1995 (left) and 2019 (right) (from Geospatial Data Cloud http://www.gscloud.cn/sources/?cdataid=263&pdataid=10).

**2 Study area**

The LRE, situated in the northwest of Liaoning Province, China (Fig. 2), is a funnel-shaped shallow estuary that exhibits a decrease in width from 20 km at the mouth to 1 km at the upstream reach. It is a momentous ecological economic zone in the north of the LDB, playing an important role in the comprehensive development of the marine industry and fishing economy in northeastern China. The bathymetry of the LRE is complicated, with the formation of

many shallow shoals due to sediment deposition. The water depth ranges from < 3 m in the shoals to < 7 m in the main


channels. The upstream river network of the LRE is relatively simple, mainly dominated by the LR and is also affected

by the river discharge from the DLR. The multi-year average river discharge of the Liao River ranges from 101 m³/s in

the dry season to 285 m³/s in wet season (Qiao et al., 2018), barely reaching 450 m³/s under extreme precipitation

conditions.

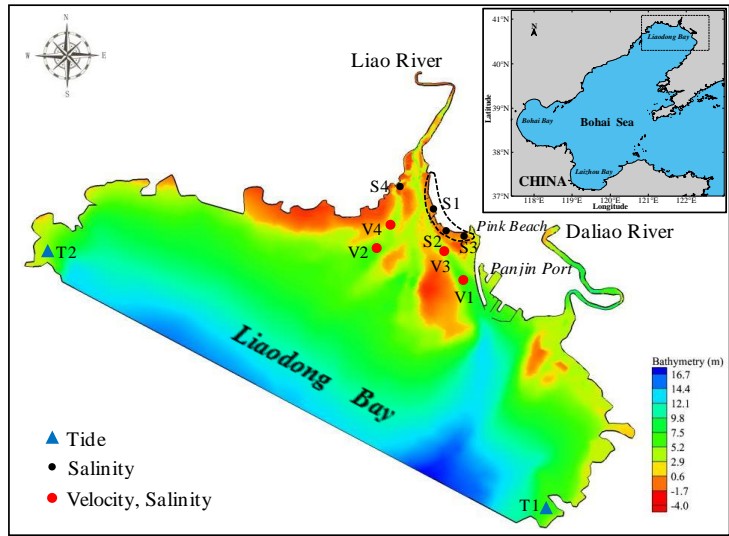

**Figure 2** The underwater bathymetry of the LRE and the location of the observed gauges. The blue triangles

denote the tidal stations, the red circles are the tidal flow and salinity observation stations, while the black circles

represent the monitoring stations used for salinity comparison.

## 3 Methods

### 3.1 Model description

FVCOM was adopted to simulate tidal flow and salinity in the LRE. This model was originally developed by Chen et

al. (2003) and improved by researchers at the University of Massachusetts-Dartmouth (UMASSD) and Woods Hole

Oceanographic Institution (WHOI) (Chen et al., 2006). Previous applications include research on large lakes, coastal

estuarine regions, and regional ocean areas (Gronewold et al., 2019; Lin and Fissel, 2014; Lai et al., 2018). The

modelling system uses an unstructured triangular grid that can accurately fit complex shorelines. The application of

terrain-following coordinates results in an improved capacity to solve complex bathymetric conditions compared to


other existing models. In addition, the ocean circulation model employs the modified level 2.5 Mellor and Yamada and

Smagorinsky turbulent closure schemes for vertical and horizontal mixing.

**3.2 Model configuration**

5       The model domain covers the whole sea region located at the top of the LDB, including the LR and the DLR,

extending from the 40.303° to 40.710° northern latitudes and the 121.029° to 122.031° eastern longitudes. In order to

visually reflect the shoreline changes resulting from the construction of the port, two coastline datasets of the LRE

were extracted from Landsat images collected in 1995 and 2019. These datasets were modified with extracted

coastlines from Google Earth, and were subsequently applied as the shoreline boundary for the modelling computation

10     domain to generate the computing grid (Fig. 3a). The bathymetry of the computation domain was based on available

DEM datasets. SMS (Surface Water Model System) software was applied to generate an unstructured triangular mesh

in the computation domain of the LRE. The total number of grid nodes and elements were 9501 and 17938 for Grid

1995 (Fig. 3b), 13699 and 25928 for Grid 2019 (Fig. 3c). The largest lateral and longitudinal lengths of the model

domain were 113 km and 67 km, respectively, while the spatial resolution of the model grids varied from 2500 m at the

15     external open boundary to 60 m at the intertidal area of the wetland domain.

The three-dimensional baroclinic mode with 11 vertical sigma levels was adopted for the modelling system, with the

internal and external mode time steps set to 2 s and 10 s, respectively. Moreover, at the open boundary condition, the

water level was determined via the validated Bohai Sea Parent Model grid at each open boundary node. The open

boundary for salinity was set to 34 PSU at the sea surface and interpolated along the sigma layers. Temperature was set

20     to 15°C across the whole domain. The initial condition for salinity was based on the steady results derived by running

the model for approximately four months. Furthermore, four case studies were set up according to different

hydrological periods (dry and wet seasons), as reported in Table 1. Following this, the river discharges of the LR and

DLR were set to 25 m$^3$/s and 48m$^3$/s (285 m$^3$/s and 266 m$^3$/s) during the dry (wet) season, respectively.

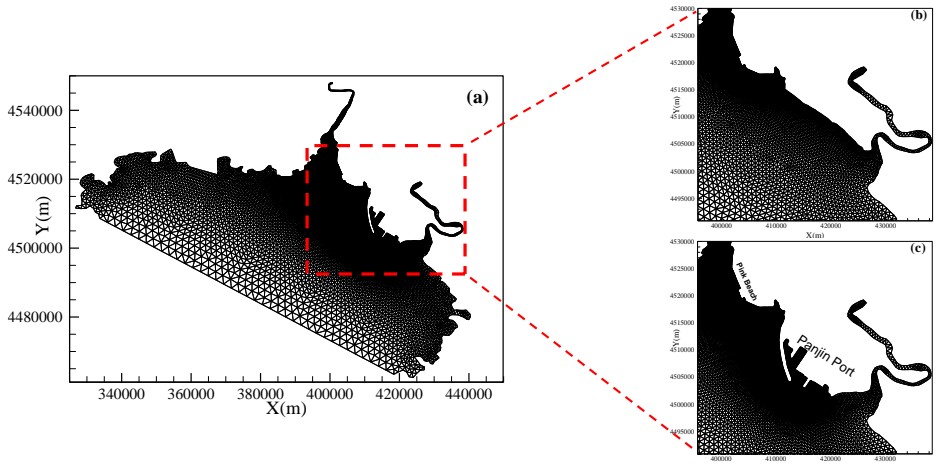

**Figure 3** Mesh sketch of the LRE **(a)** and detailed meshes for the Panjin Port region on Grid 1995 **(b)** and Grid 2019

**(c)**.

**Table 1** Model case study settings.

| Case Study | Grid | River Discharge (m³/s) | |
|---|---|---|---|
| | | LR | DLR |
| Case1 | 2019 | 25 | 48 |
| Case2 | 1995 | 25 | 48 |
| Case3 | 2019 | 285 | 266 |
| Case4 | 1995 | 285 | 266 |

**3.3 Model validation**

To order to evaluate the model's performance in the simulation of the tidal dynamics and salt transport in LRE, the

simulated results were compared with the available measured data. In particular, two tide level stations (T1 and T2),

four tidal flow measuring sites (V1, V2, V3 and V4), two salinity measuring sites (V1 and V2) and four wetland

10   salinity reference sites (S1, S2, S3 and S4) were selected to compare and analyze the model performance in the study

domain. The model simulations cover the period from 20 April to 10 September 2018. Comprehensive model

validation was performed using the observation data. The tide level was verified from 0:00 on 1st May to 23:00 on 31

May 2018. The periods of tidal flow validation were divided into neap tide and spring tide, of which the neap tide

period was from 11:00 on 8 June to 10:00 on 9 June, 2018, and the spring tide period from 6:00 on 15 June to 5:00 on

15   16 June, 2018. The salinity data was measured at V1 and V2, with a validation period from 5:00 on 10 September to

4:00 on 11 September 2018 for V1, and from 6:00 on 10 September to 5:00 on 11 September 2018 for V2.





The comparison between modelled and observed data was quantified using the root mean square error (RMSE) and the predictive skill score (Skill). The magnitude of the RMSE indicates the average deviation between the simulated results and the observed data, while the predictive skill score represents the degree to which the simulated value fits the measured value (1 = perfect fit, 0 = complete difference). The RMSE and Skill are defined as follows:

$$RMSE = \sqrt{\frac{\sum_{i=1}^{n}(M_i - O_i)^2}{n}} \tag{1}$$

$$Skill = 1 - \frac{\sum_{i=1}^{n}(M_i - O_i)^2}{\sum_{i=1}^{n}(|M_i - \bar{O}| + |O_i - \bar{O}|)^2} \tag{2}$$

where $M_i$ and $O_i$ represent the modeled results and observed values, respectively, $\bar{O}$ indicates the mean of the observed values, and $n$ is the number of observed data points.

Figure 4 presents the tide level comparison results. The simulated water levels at T1 and T2 are observed to agree well

with the measured data. In general, the amplitudes of the modeled results are consistent with the observations, with a maximum phase difference of less than 30 minutes. Despite this, several significant errors are observed between the simulated high and low tide levels and the observed values. This may be attributed to the inaccurate open boundary water level conditions. The comparisons of the simulated and measured flow speeds and directions during neap and spring tides are presented in Figs. 5 and Figure 6, respectively. In general, the modeled tidal flow exhibits a better

match with the observation data in neap tide compared to spring tide. The poorer spring tide fitting results may be a result of the following: (1) The open boundary conditions adopted in the model may be inaccurate for certain periods; (2) the bathymetry used in the model may differ to that of the study area due to recent sediment deposition; and (3) the model employed a uniform bed roughness coefficient across the whole area. Figure 7 displays the comparisons of simulated and observed salinity at V1 and V2. The simulated salinity is in good agreement with the measured data,

with broadly consistent trends in variation between datasets. Table 2 reports the results from the error quantification analysis of the simulated tide level ($\zeta$), tidal flow ($\bar{U}$) and salinity ($\bar{S}$). The Skill scores for the T1 and T2 tide levels are both greater than 0.95, while flow velocities exceed 0.80. The results indicate that the model established in this paper can accurately reproduce the hydrodynamic processes in the LRE. The determined Skill values for salinity





indicate that the model effectively reproduces the salinity variations at V1, yet slightly under-estimates the salinity at

V2. The high simulation accuracies of the water level, currents and salinity determined by the proposed model in the

study area demonstrates that the model is suitable for further studies on the salinity distribution in the LRE.

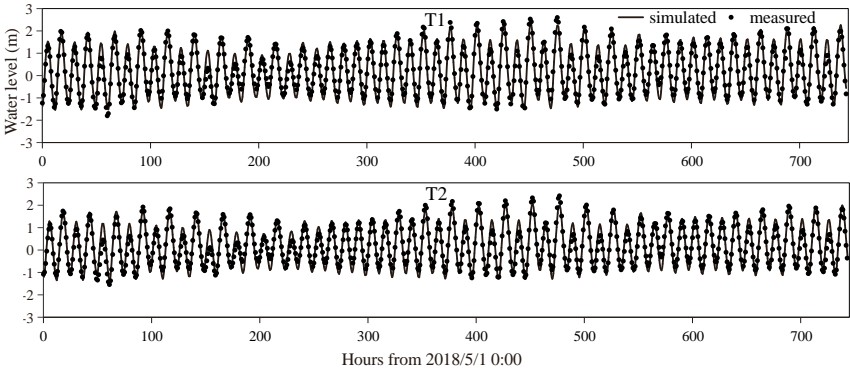

5    **Figure 4** Comparison of the simulated and measured water levels at T1 and T2.

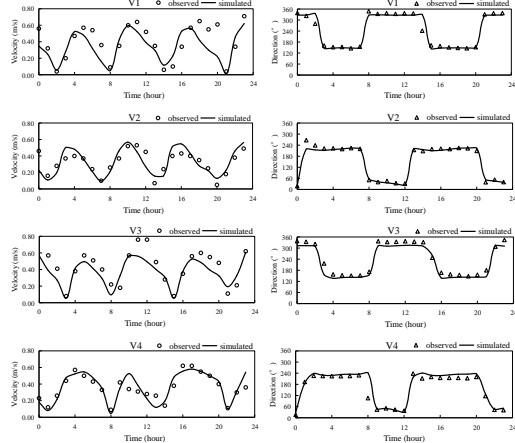

**Figure 5** Comparison of simulated and measured tidal currents at the monitoring stations during neap tide.





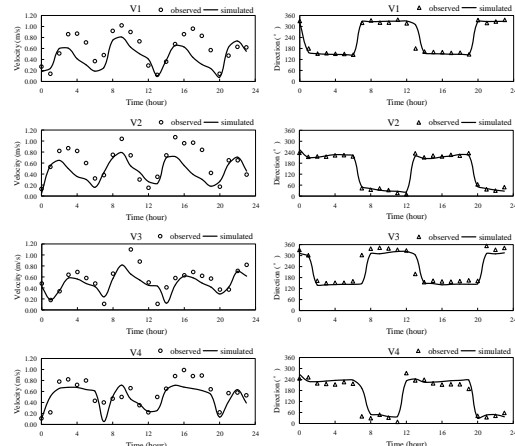

**Figure 6** The comparison of simulated and measured tidal currents at the monitoring stations during spring tide.

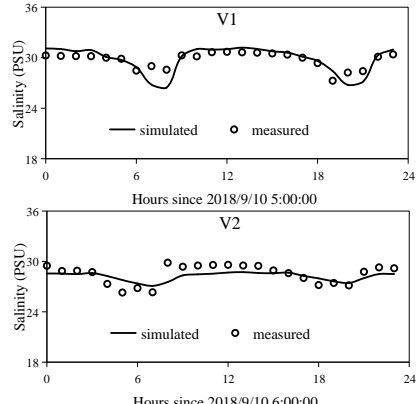

5    **Figure 7** Validation of salinity at (a) V1 and (b) V2.

**Table 2** Error analysis of simulated and measured values.

| Stations | RMSE | | | Skill | | |
|---|---|---|---|---|---|---|
| | $\zeta$(m) | $\bar{U}$(m/s) | $\bar{S}$(psu) | $\zeta$(m) | $\bar{U}$(m/s) | $\bar{S}$(psu) |
| T1 | 0.248 | — | — | 0.985 | — | — |
| T2 | 0.305 | — | — | 0.969 | — | — |
| V1 | — | 0.207 | 0.888 | — | 0.808 | 0.859 |
| V2 | — | 0.196 | 0.877 | — | 0.809 | 0.701 |
| V3 | — | 0.160 | — | — | 0.849 | — |
| V4 | — | 0.137 | — | — | 0.891 | — |

**4 Results and discussion**



### 4.1 Effect of river discharge on salinity distribution in the LRE

Due to the recent rapid urban sprawl and growth in population, water consumption has increased dramatically in the LR basin. In addition, the upstream closure projects have directly led to a sharp decrease in the runoff of the LR, which may further affect the exchange and mixing of water bodies in the estuary (Ralston et al, 2008; Gong et al, 2011). In

order to explore the effect of river discharge on salinity transport in the LRE, two numerical cases representing the dry season and wet season were selected (Case1 and Case3 in Table 1). The 50-h averaged salinity distributions of the LRE during spring and neap tides under different runoff conditions are shown in Figure 8 and 9. During the dry season, the salinity in the middle of the LRE during neap and spring tides is maintained at 16-28 PSU, while the isohaline is relatively sparse from downstream to upstream. During the wet season, the salinity value of the middle LRE deceases

significantly and is maintained at 4-20 PSU. Moreover, the density of the isohaline in the estuary increases from head to mouth. As the tidal current movement strengthens during the spring tide period, the saltwater intrusion distance in the estuary increases compared to that during the neap tide period. The salinity distribution characteristics at the surface and bottom layers are generally consistent due to the shallow water depth in the LRE. In order to quantify the effect of river discharge on the salinity of LRE, the 30-day averaged salinities of eight stations (Fig. 1) at the surface

and bottom layers during the dry season (Case1) and wet season (Case3) were compared (Fig. 10). The salinity of the LRE is observed to decrease dramatically as the river discharge increases. Taking Figure 10 as an example, the averaged surface salinities of the eight sites (S1, S2, S3, S4, V1, V2, V3 and V4) during the wet season decreased by 85.1%, 28.8%, 14.5%, 68.3%, 18.7%, 40.7%, 50.2% and 47.4%, respectively, compared with those in the dry season. In addition, there is an inverse relationship between the distance from the head of the estuary and the decrease in

salinity. As a result, the salinity of the LRE varies with the season, with river discharge changes exerting a significant effect on the longitudinal distribution of salinity in the estuary. This indicates the important effect of salinity variations on changes in marsh vegetation. Due to the construction of closure projects in the upper reaches of the LR, the emptying of freshwater discharge from the LR into the LDB decreases sharply, directly affecting the variation of the salinity gradient in the estuary. Thus, the increasing salinity will inhibit the growth of *S.h.* and *P.a.* vegetation in this



estuarine wetland, which could potentially lead to the degradation of estuarine wetland communities and the

destruction of estuary ecosystems (Popescu et al. 2015).

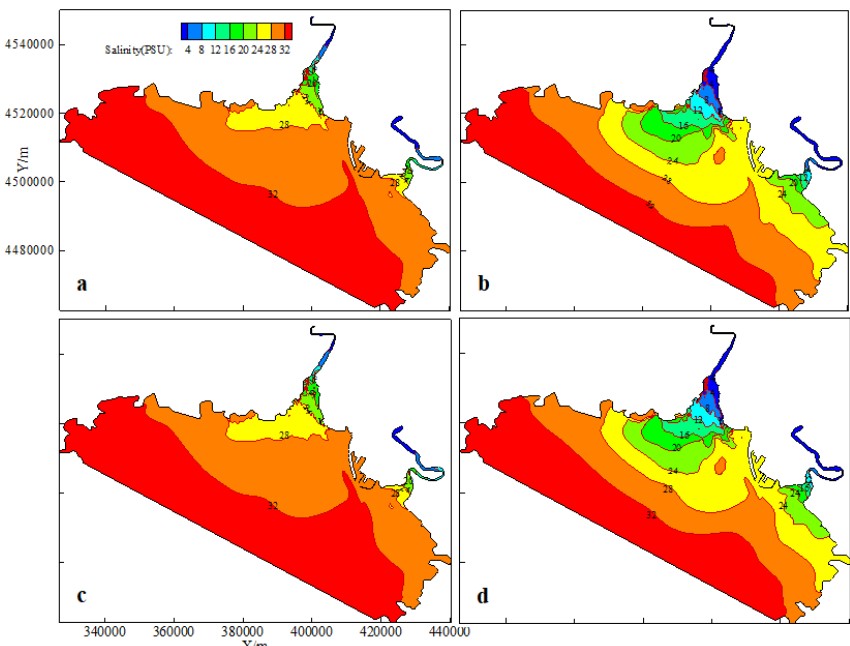

**Figure 8** 50-h averaged salinity at the (a)-(b) surface and (c)-(d) bottom during spring tide for Case1 (left panel) and

5  Case3 (right panel).

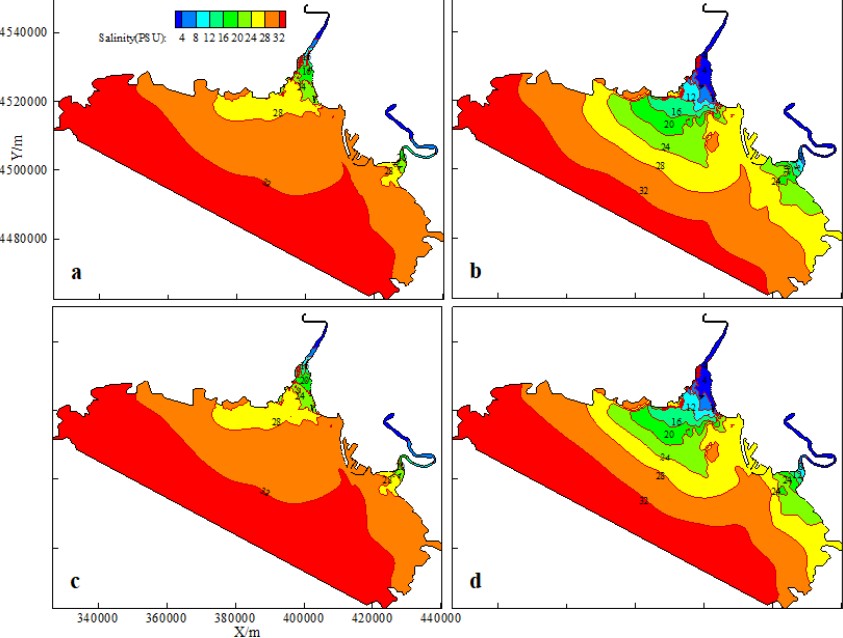

**Figure 9** 50-h averaged salinity at the (a)-(b) surface and (c)-(d) bottom during neap tide for Case1 (left panel) and





Case3 (right panel).

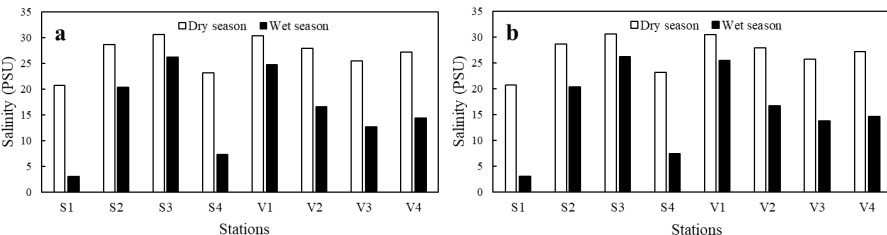

**Figure 10** 30-day averaged salinity at the (a) surface and (b) bottom of the eight monitoring sites during the dry and
wet seasons.

## 4.2 Effect of shoreline change on salinity distribution in the LRE

Panjin Port is located on the southeast of the LRE, adjacent to the DLR (Fig. 2). The port was founded in 1995,

completed in 1999, and following more than 10 years of continuous expansion, the existing land area has reached

230,000 m$^2$. The construction of Panjin Port is of great importance for the development of the local economy, yet its

construction has also been problematic. The large area of marine reclamation has resulted in drastic changes in the

southeast coast of the LRE. Severe degradation has occurred in recent years across the wetlands of the LRE,

particularly in the Pink Beach Wetland (PBW) close to the northwest of Panjin Port. The majority of the vegetation in

the PBW grows on the tidal flats along the coast, and is thus submerged by sea water at high tide and is exposed at low

tide. This unique growth environment complicates the research on the degradation mechanisms facing such vegetation.

Thus, we applied numerical simulations to overcome this obstacle, focusing on the influence of shoreline changes

caused by anthropogenic activity on the salinity distribution of LRE. In addition, we further discussed the potential

relationship between salinity changes and estuarine wetland degradation. In order to explore the influence of shoreline

changes caused by the port construction on the salinity distribution in the LRE, the computational grids generated by

the two shorelines in 1995 (pre-port construction) and 2019 (post-port construction) were adopted for salinity

distribution simulations in the LRE. The simulated results of the two grids were then compared and analyzed. The

average salinity distribution calculated by the two grids during the dry and wet seasons are illustrated in Figs. 11 and

12, respectively. The average downstream salinity at the PBW simulated by Grid 1995 lies within the range of 26-28

PSU, while that of Grid 2019 has a range of 28-30 PSU (Fig. 11). The salinity of the estuary exhibits a significant

decrease during wet season, with the longitudinal salinity distribution gradient becoming more pronounced (Fig.12).

The average downstream salinity at the PBW of the 1995 shoreline ranges between 20-22 PSU, while that of the 2019

lies within 24-26 PSU. For a more intuitive comparison, the isohalines in the LRE during the dry and wet seasons were

5    plotted using the results from the two grids, as shown in Figure 12. The green isohalines (grid 2019) are located above

the red isohalines (grid 1995) in the estuary waters during both seasons, particularly in the eastern region of

downstream PBW. This indicates the intensification of the shoreline change with the intrusion of salt water.

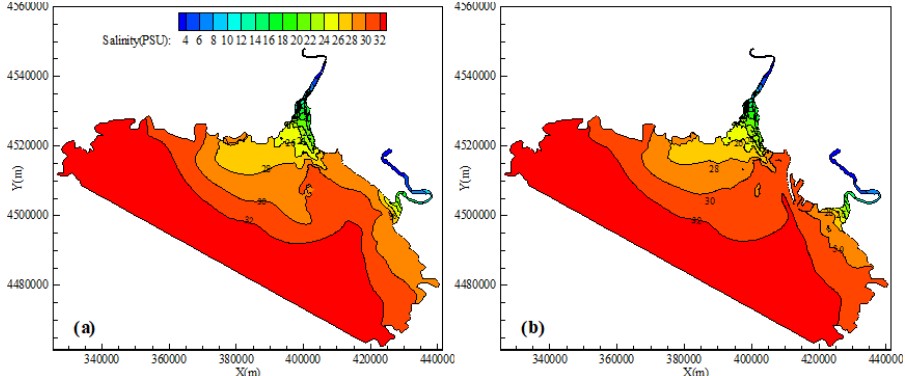

**Figure 11.** Averaged salinity distribution for (a) grid 1995 and (b) grid 2019 during the dry season.

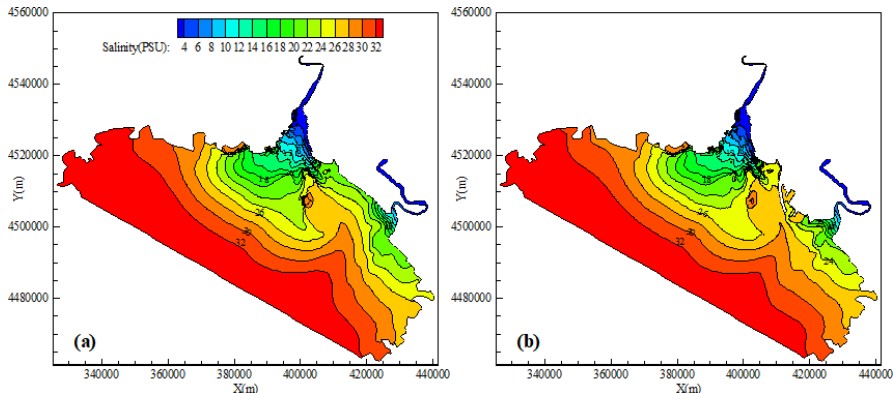

**Figure 12** Averaged salinity distribution for (a) grid 1995 and (b) grid 2019 during the wet season.

In order to quantify the influence of shoreline changes on the salinity of the LRE, we compared the 30-day averaged

salinities of the surface and bottom layers determined via the two grids at the eight monitoring sites (Fig. 13). During

15   the dry season, following the construction of the Panjin Port, the averaged salinities of the eight stations (S1, S2, S3,


S4, V1, V2, V3 and V4) increased by 4.4%, 6.2%, 7.9%, 0.7%, 4.5%, 2.6%, 5.1%, 2.6%, 5.1% and 2.3%, respectively.

During the wet season, the averaged salinities subsequently increased by -0.6%, 14.5%, 20.9%, -1.3%, 6.9%, 2.2%, 11.0% and 1.4%, respectively. These results indicate that the shoreline changes caused by the extensive sea reclamation for the construction of Panjin Port have a noticeable effect on the salinity distribution in the LRE. In

5    particular, the change of salinity in PBW (S2, S3) with the closest distance to Panjin Port is the most significant. The salinity of the waters downstream to PBW increases following the construction of the port. This increase is observed to be greater in the wet season compared to the dry season, with a maximum increase in salinity of 4 PSU.

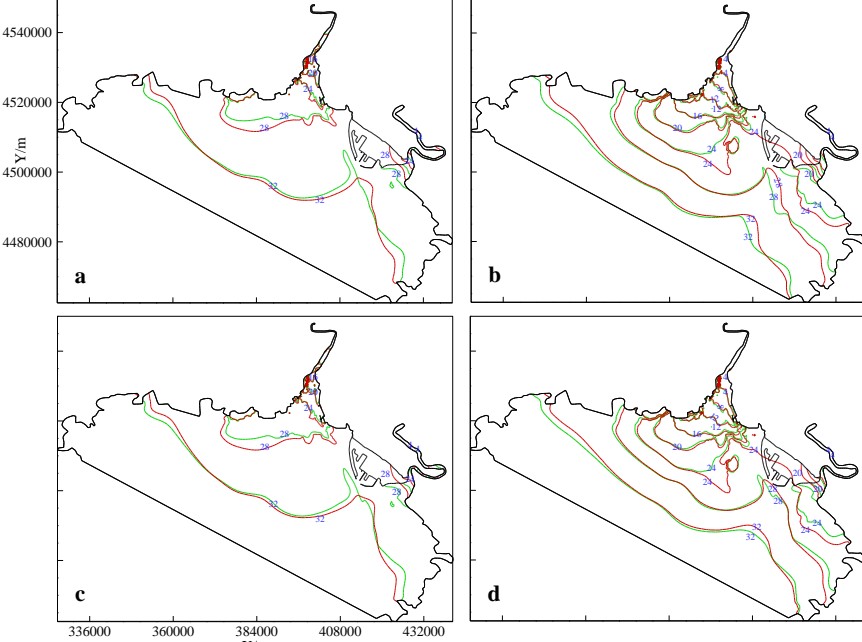

**Figure 13** Averaged salinity for grid 2019 (green solid line) and grid 1995 (red solid line) at the (a)-(b) surface layer
10    and (c)-(d) bottom layer during the dry season (left panel) and wet season (right panel).





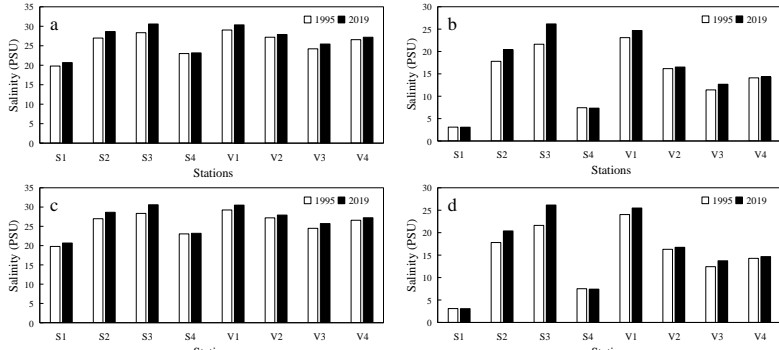

**Figure 14** 30-day averaged salinities of the eight monitoring stations derived from the 1995 and 2019 grids for the (a)-(b) dry season and (c)-(d) wet season.

### 4.3 Effect of shoreline change on tidal flow patterns in the LRE

Large areas of marine reclamation can alter both the local topography and the original coastline (Jia et al., 2018).

Changes in the shoreline will have an impact on the characteristics of the original flow field, which can consequently

affect the transport processes of ocean materials, including sediment transport, pollutant diffusion and salinity

distribution (Alebregtse and de Swart, 2016). To investigate the relationship between the flow field and salinity

variation as a function of shoreline changes due to reclamation, we analyzed the tidal flow and residual flow fields of

the LRE with and without the Panjin Port. The results were then used to illustrate the underlying mechanisms behind

variations in salinity in terms of mass transport. Comparisons of the residual flow fields of the Liao River under the

two shoreline conditions (Fig. 15) demonstrate that at the 1995 shoreline, a residual flow induced by the runoff from

the DLR traveled along the coastline downstream of the PBW, diluting the sea water due to the constant replenishment

of fresh water. Following its establishment, the residual current to the south of the port exhibited significant changes in

both its magnitude and direction, with an eastward flow that subsequently moved southeastwards. A residual was

observed around the tip of the LRE with a strong (approximately 20 cm/s) clockwise circulation close to the Panjin

Port, resulting from a flood-dominant north and ebb-dominant south shore. The spatial distribution of the residual

currents exhibited a similar trend during both neap and spring tides. The obstruction of the port body and enhanced

tidal mixing effect resulted in the diffusion of the DLR runoff, weakening the fresh water dilution effect on the



seawater. This can explain the higher salinity in the PBW following the shoreline changes in 2019. In summary, with the exception of the upstream river discharge, the construction of Panjin Port has had a marked effect on the variation of salinity in the LRE, with impacts including the potential degradation of wetlands in the Liao River Estuary, particularly the Pink Beach Wetland.

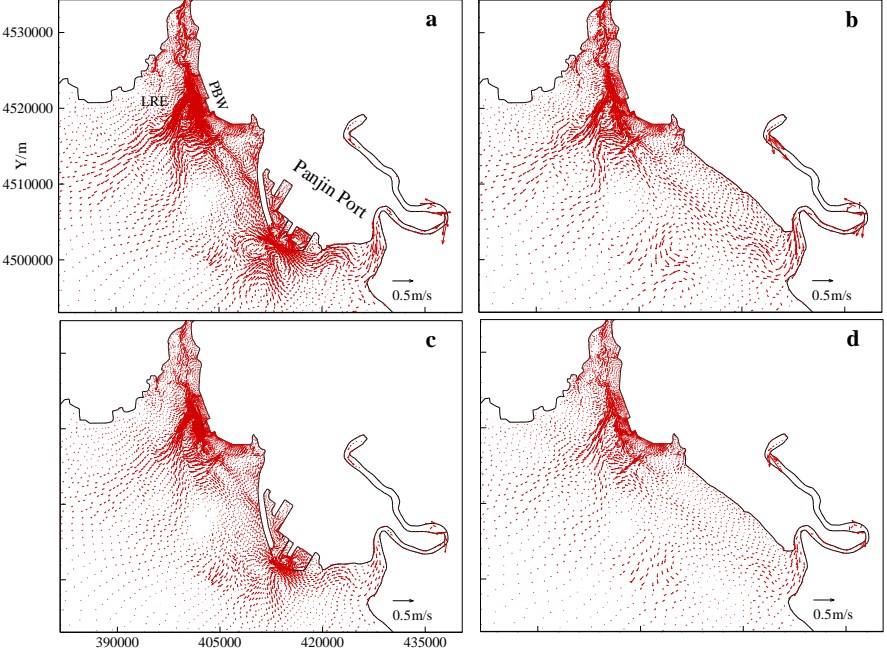

**Figure 15** Residual flow at the (a)-(b) surface layer and (c)-(d) bottom layer for Grid 2019 (left panel) and Grid 1995 (right panel) during the dry season.

**5 Conclusions**

10    In this paper, a well-validated three-dimensional numerical model was applied to investigate the influences of river discharge and shoreline changes on the salinity distribution in the LRE. Two runoff conditions were set to represent different water seasons in the estuary. Data derived from Landsat satellite imagery and adapted using Google Earth software was employed to extract the LRE shorelines in 1995 and 2019. These were then used to generate the computational grids for tidal current and salinity distribution simulations of the LRE before and after the construction of Panjin Port. Simulation results indicate that the variation of river discharge has a significant effect on the

longitudinal distribution of salinity in the LRE. Runoff increases can significantly decrease salinity in estuary waters

due to the dilution of freshwater, while variations in salinity are observed to be negatively correlated with the distance

from the upper reaches of the estuary. Furthermore, the changes in shoreline triggered by the construction of Panjin

Port have a marked effect on the salinity transport in the LRE, with the Pink Beach Wetland in northwest Panjin Port

observed to be the most effected. There is a rise in the water salinity amongst the wetlands located within the lower

reaches of the Pink Beach Wetland following the construction of the port. The rate of increase is greater during the wet

season compared to the dry season, with a maximum increase in salinity of 4 PSU. Comparing the residual flow field

simulated using the two shoreline conditions reveals that prior to the port construction, the runoff-induced residual

flow from the DLR travelled along the shoreline to the lower reaches of the Pink Beach Wetland, where it then

accumulated. This continuous supply of fresh water diluted the sea water and reduced the salinity. Once the port was

constructed, its obstruction of the port area strengthened the tidal current mixing, and the partial runoff from the DLR

was diverted. As a result, the fresh water dilution effect further weakened, thus increasing the salinity of the sea water

in the lower reaches of the Pink Beach Wetland.

The ecological degradation of wetlands in the LRE has become more and more severe in the past decade. This is

attributed to long-term disturbances resulting from anthropogenic activity, including river closure projects, agriculture

irrigation, and coastal constructions. Despite this, key physical processes in the LRE have yet to be systematically

investigated and understood in terms of river discharge, wind, wave and complex terrain conditions. The influences of

wind stress, estuary topography variations and water resource utilization in upper rivers on the spatial-temporal

distribution of salinity in the LRE will be the focus of future work.

**Author contributions.** All authors contributed to the design and development of the work. The experiments were

originally carried out by M Z and T X. H J carried out the data analysis. M Z and T X built the model and wrote the

paper.

**Competing interests.** The authors declare that they have no conflict of interest.



**Acknowledgements.** This work was supported by the National Key R&D Program of China (2019YFC1407704), the

National Nature Science Foundation of China (51779039, 51879028).

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
