# Peer review of "The impact of runoff decrease and shoreline change on the salinity distribution in the wetlands of"

_Ocean Science, 2020_

## Referee Comment (RC1) · William Pringle (Referee) · 12 Sep 2020

General Comments:

This study uses a 3D hydrodynamic model to investigate the impacts on salinity distribution in the wetlands of the Liao River Estuary (LRE) due to the construction of Panjin Port under wet and dry river discharge conditions. The study shows that the port construction prevents fresh water from the Daliao River discharge being transported up the coastline to Pink Beach Wetland, thus increasing the salinity. The implication being that this increasing salinity could be detrimental to the growth of vegetation (Suaeda heteroptera (S.h.) and Phragmites australis (P.a.)) in the LRE wetlands.

[Figure]

I think that the results are fairly well presented and make sense scientifically, and the study broadly achieves the principle aims that are clearly stated in the introduction. So, I think it should be considered for publication after addressing my comments to follow.

The authors could be more careful and precise about the current state-of-the-art knowledge on the effects of salinity on S.h. and P.a. vegetation. Do we know the levels of salinity that might be expected to be troublesome? i.e., does the increase to salinity from the port construction actually make the salinity levels reach a point that is expected to affect S.h. and P.a.? I also think it would help to show a satellite image (or any other image) that illustrates the degradation of the LRE wetlands seen over the past decade that is described by the authors.

The method section contains some inaccurate statements and many parts are poorly described, so it needs careful attention to improve it. The authors can refer to my specific comments for details. Nevertheless, the model validation results appear acceptable to me.

Lastly, the results sections start by including details that really should be in the introduction and the methods section (and are in the past tense). I think it would help to move all these details from the results sections to bolster the introduction and methods sections.

Specific Comments:

1. The authors are very liberal with their use of acronyms making some parts hard to follow. Only need to use acronym if the word is repeated many times and is long. I recommend to just use an acronym for the Liao River Estuary (LRE) and the vegetation (P.a. and S.h.), and spell everything else out.

2. Page 3, Line 14: What types of models are "ELCIRC and COAWST"?

3. Page 4, Lines 5-8: "Estuarine salinity has a significant effect on the growth of coastal wetland plants plays an important role in maintaining the ecological health of estuarine

wetlands. Despite this, studies on the spatial and temporal distribution of salinity in estuarine wetlands are limited, with most work generally focusing on salinity transport and saltwater intrusion mechanisms along the estuaries". These two sentences seem to come out of nowhere and do not have any references to back up the statements. Please expand on these sentences and provide details of the previous studies and their limitations.

4. Page 4, Lines 11-12: "According to satellite images and on-site surveys, the wetlands in the LRE have experienced severe degradation over the past decade". What surveys and satellite images are these? Are these from the authors' own studies? Provide more details.

5. Page 4, Lines 14-15: "Studies have revealed that salinity increases in water and soil can result in the death of S.h. vegetation." What studies are these? Please provide references. Furthermore, what about P.a. vegetation? No details on P.a. are provided in this paragraph.

6. Page 5, Line 4: "(2) explore the internal mechanisms of these effects". What does internal mechanisms mean here exactly?

7. Page 6, Lines 16-17: "The application of terrain-following coordinates results in an improved capacity to solve complex bathymetric conditions compared to other existing models." This is not a correct statement. Many other models use terrain-following coordinates (e.g., ADCIRC, SELFE/SCHISM, ROMS) and they have a well-known issue associated the computation of the pressure gradient term in high gradient regions (Haney, 1990) that other researchers have attempted to alleviate (e.g., SELFE/SCHISM uses hybrid coordinates (Zhang et al., 2015)).

8. Page 7, Lines 6-10. How did the coastlines from Google Earth differ from the coastlines from Landsat Images? Please explain in more detail about the coastline extraction process (what tools?). Where were the Landsat images used and where were the Google Earth images used?

9. Page 7, Lines 10-11. "available DEM datasets" does not explain anything. Provide the source.

10. Page 7, Lines 14-15. Does the spatial resolution model grid vary only with distance between the open boundary and the wetland? Or is there some bathymetry depth function involved as well?

11. Page 7, Lines 16-17. Says the internal mode time step is 2 s and external mode time step is 10 s. I think this should this be reversed. (the external mode is the fast barotropic mode and should have a smaller time step).

12. Page 7, Line 18. What is the "validated Bohai Sea Parent Model grid"? Any reference? Explain in more detail please.

13. Page 7, lines 18-19. What do you mean by "salinity was set to 34 PSU at the sea surface and interpolated along the sigma layers"? Interpolated between 34 PSU and what other value?

14. Page 8, Lines 12-13 & Line 16. Are the open boundary water level conditions inaccurate? Why open boundary conditions may be inaccurate for certain periods? You should be able to quantify this from the "validated Bohai Sea Parent Model grid".

15. Page 12, Lines 11-12. I don't really see that much evidence in Figures 8 and 9 that "during the spring tide period, saltwater intrusion distance in the estuary increases compared to that during the neap tide period." Can you be more specific about where you see that? Including a panel showing the differences may help.

16. Page 12, Line 24 – Page 13 Lines 1-2. It's unclear to me how the Popescu et al. (2015) reference relates to this sentence. I searched for keywords "salinity" and "salt" in their article and nothing comes up. Furthermore, this sentence implies that increasing salinity definitely inhibits growth of S.h. and P.a. vegetation, but the introduction on Page 4 lines 10-11 says that the "S.h and P.a. are the most common pioneer salt-tolerant plant in the LRE wetlands". Of course this does not mean that a very high level

of salinity can't inhibit their growth, but this ties back to my earlier specific comment 5) where I think you need to be more clear and careful about the statements relating to how both S.h. and P.a. are thought to be influenced by salinity, and make the correct citations.

17. Most of the beginning of Section 4.2 (page 14) should be in the introduction and methods section. The first ∼5 lines of Section 4.1 and 4.3 are the same. Please only focus on including results in Section 4.

18. Page 15 Line 7: "This indicates the intensification of the shoreline change with the intrusion of salt water". This needs to be reworded, do you mean that the shoreline change increases the intrusion of salt water onto the wetlands?

19. Page 17, Line 10: says both tidal flow and residual flow were analyzed but only results for the residual flow are presented.

20. It would be nice to use colormaps that are more physically intuitive and unbiased instead of the rainbow ones adopted; refer to Thyng et al. (2016) for colors design to be used for salinity and depths etc.

Technical Corrections:

1. Page 15 Line 5: Figure 12 should be Figure 13.

2. Page 15, Line 14: Fig. 13 should be Fig. 14.

References

Haney, R.L., 1990. On the Pressure Gradient Force over Steep Topography in Sigma Coordinate Ocean Models. J. Phys. Oceanogr. 21, 610–619.

Thyng, K.M., Greene, C.A., Hetland, R.D., Zimmerle, H.M., DiMarco, S.F., 2016. True colors of oceanography: Guidelines for effective and accurate colormap selection. Oceanography 29, 9–13. doi:10.5670/oceanog.2016.66

Zhang, Y.J., Ateljevich, E., Yu, H.C., Wu, C.H., Yu, J.C.S., 2015. A new vertical coordinate system for a 3D unstructured-grid model. Ocean Model. 85, 16–31. doi:10.1016/j.ocemod.2014.10.003

---

## Referee Comment (RC2) · Anonymous Referee #2 · 15 Sep 2020

General comments

This paper applies the FVCOM unstructured mesh model to the Liao River Estuary. The aim of the study is to understand how the construction of the Panjin Port impacts the flow and salt intrusion in the estuary. FVCOM is used to simulate hydrodynamics and salinity, under different river discharge scenarios. The model is validated using recent observations in intertidal areas. Results show that the construction of the port has a large impact on salinity transport in the Lao River Estuary. The Pink Beach wetland is the most affected area with a rise in water salinity due to modification of the water flow. This results from the construction of the port that prevents fresh water from

the Daliao River to be transported up to the lower reaches of the Pink Beach wetland. This is an important result as this increase in salinity has an impact on vegetation growth in the area.

The paper is well written on the whole, and provides readers with useful information about wetland hydrodynamics. However, some parts of the manuscript should be improved prior to publication. In particular, there is not enough information about the hydrodynamics of the study area and reference to previous studies in the introduction, the model is poorly described and figure 15 has to be reworked.

Specific comments

Introduction:

Information about the main circulation features and tidal dynamics of the coastal Liaodong Bay from the literature would be nice (for example the tide is semi-diurnal close to the coast according to Hao et al., 2005).

But my main concern is that there is nearly no mention to the paper by Qiao et al. (2018), entitled "Numerical study of hydrodynamic and salinity transport process in Pink Beach wetlands of Liao River Estuary, China", which shares co-authors with this paper. In their paper, Qiao et al. apply the Mike unstructured mesh model to the Liao River Estuary. They focus on the hydrodynamic characteristics and salinity transport processes in Pink Beach wetland of the Liao River Estuary, considering the effect of wetland plant on tidal flow. In the present paper, the authors should emphasize what is new in their study (scenarios with and without the port). Why do they use FVCOM instead of MIKE, is there a reason? In their conclusion, the authors of the present paper mention (Page 19) that runoff increases can decrease salinity in estuary waters due to the dilution of freshwater. Is this result really new? In Qiao et al., Figure 20 shows contour maps of salinity in the LRE under different runoffs during the period of highest saltwater intrusion, and the authors conclude that "the larger the river discharge, the stronger the runoff diluting effect", and "when the river discharge is low, less freshwater

is mixed into the system and salinity is higher". Page 4 Line 20 : Sources of data and where they can be downloaded should not be in the introduction but in the Method part.

Model description:

It would be nice to have more details about the model equations and how they are solved. At least mention that it is a 3D primitive equation model.

Page 6 Line 17: please add details about the vertical coordinates (sigma?) and provide reference.

Page 7 Line 1: "other existing models". As many other models use terrain following (sigma) coordinates, what do you mean? Is it models that use z coordinates? You should be more precise here.

Model configuration:

Model initialization and forcing need to be more detailed. Please give the model initial condition and the boundary conditions for the tides, or add a reference. Also, explain your choices for open boundary salinity of 34 PSU, initial temperature of 15°C, and river discharge scenarios (are values chosen from observation, literature?).

Page 7 Line 11: Please provide a reference for surface water model system. As you do not use the acronym SMS after, do not use it here.

Page 7 Line 18: A reference for the Bohai Sea Parent Model validation would be nice.

Model validation:

Page 9 Line 4: You could add "by taking into account the bias and correlation between model and observation" to the description of skill sentence.

Page 9 Line 11: you claim that "significant errors are observed between the simulated high and low tide levels and observed values". If this is from figure 4, it is not very clear to me. Also, T1 and T2 are very close to the boundary, what is the tidal forcing

at the boundary? You explain the poorer fitting results at spring tide by the choice of open boundary conditions, so you may definitely give them in the model description/configuration part. It would be great to have this kind of comparison close to the LR, can we assume that there is no data there?

Results and discussion:

Page 12 Line 6: Why did you choose 50 hours for averaging?

Page 17 Line 13-19: Figure 15 is not clear and should be reworked, as this part is not very clear for the moment.

Page 18 Line 1-4: As this part deals with the effect of shoreline change on tidal flow, all the text beginning by "In summary" could be moved to the conclusion, or the link with salinity could be added in the subtitle.

Conclusions:

Page 18 Line 10: what is a well-validated model? Maybe you could use a term that refers to the robustness of the model ("proven model"?).

Figures:

Figures 1 and 2 could be merged to give the location of the area at first, and the names of big cities could be added (at least Dalian) to facilitate the reading by foreign scientists.

Figure 2: Blue triangles are not visible on the plot. Please add a Table with the coordinates of the stations.

Figure 4: It would be nice to have the amplitude and phase for the main tidal components for the comparison in an additional Table or in the text.

Figure 6: Why did you choose the scale 18-24 PSU? Is it possible to zoom in?

Figures 8 and 9: Is it necessary to show both surface and bottom maps, as they look

very similar? You do not comment the differences in the text so I suggest to remove bottom plots.

Figure 15: This figure has to be reworked, as it is very hard to see anything, especially the direction of arrows. Perhaps reducing the number of arrows and zooming in areas of interest could help?

Minor comments

Page 3 Line 14: What are ELCIRC and COAWST? Models?

Page 7 Line 11: Please write "digital elevation model" instead of DEM.

Page 7 Line 12: "elements were respectively"

Page 8 Line 11-12: Delete Âń comprehensive model validation was performed using the observation data", as it has already been said at Line 8.

Page 12 Line 23: Please write "Liaodong Bay" instead of LDB.

Page 15 Line 5: I suggest to replace "above" with "upstream".

Page 15 Line 5: Replace "Figure 12" by "Figure 13".

Page 17 Line 15: By "its", do you mean "the port"?

Page 17 Line 19: Add "(not shown)" for the comparison of results at neap and spring tides.

Typos:

Page 2 Line 21: "to understand"

Page 8 Line 7: "In order to"

Page 9 Line 14: "Figure 5 and Figure 6"

Page 12 Line 7: "Figures"

Page 12 Line 9: "decreases"

---

## Referee Comment (RC3) · Anonymous Referee #3 · 24 Sep 2020

General comments

This manuscript describes a series of FVCOM experiments that aim to investigate the sensitivity of salinity distribution in Liao River Estuary to the construction of the Panjin Port. Motivation is driven in part by considering that the survivability of two species of salt-tolerant plants in the LRE may be negatively affected.

The model setup section is in need of an overhaul to better describe what has been done here (details below), and more broadly the manuscript should receive some re-organization to get all of the right content into the right sections. For example the start of section 4.1 has introduction material (pg 12, lines 1-5), and section 4.2 also has in-

troduction material (pg 14, lines 8-18), while at pg 14, lines 18-21 look like they belong in a methods section.

Despite the organizational issues, the manuscript starts off with a focused goal of investigating salinity distribution sensitivity and demonstrates/quantifies the effects of adjusting the domain geometry. In that regard it is successful. Surface forcing is absent so there is a lack of completeless for the model, but this is noted for future work, and presumably explains why specific forecasts about the fate of the salt-tolerant plants is not made.

Specific comments

pg 7, lines 17-18: What is the "validated Bohai Sea Parent Model"? Definitely we need some sort of citation here, and some rationale for using it's water level as OBC in the present study. Does this model provide tides only or does it also include non-tidal sea-level variability?

pg 7, lines 18-19: "The open boundary for salinity was set to 34 PSU at the sea surface and interpolated along the sigma layers." I don't follow; more than one number is needed to conduct an interpolation. Does this mean the salinity OBC was set to 34 PSU at all depths along the entire open boundary? Is the 34 PSU isohaline known to coincide with the model open boundary? How does that reconcile with estuarine flow characterised by salty inflow at depth and fresher outflow in upper layers?

pg 7, lines 19-20: "Temperature was set to 15°C across the whole domain." This sounds like the temperature initial condition. What was done for the temperature OBC? Was temperature an active or inactive tracer in the model? Is water temp mostly uniform in the area justifying setting it to inactive in the model?

pg 7, lines 20-21: "The initial condition for salinity was based on the steady results derived by running the model for approximately four months." This reads as if the final conditions were used to initialize the model! Presumably the model was started with a

different set of conditions (perhaps uniform salinity?) and then the four month run used (perhaps with average runoff?) to produce quasi steady salinity field for initializing the four cases. Please clarify this.

pg 7: No information provided about velocity boundary conditions.

pg 7: No justification for not including surface forcing.

pg 7, lines 21-23: How were these numbers selected? Are there discharge records that were used? pg6 line 3 mentions low average and high averages of 101 & 285 m3/s for LR but a value of 25 is used for cases 1&2. Some rationale for the choices would help here.

pg 8, lines 12-15: It is typical to evaluate water level by splitting into tidal and residual/sub-tidal components, this would help in understanding if mismatches are due to to poorly tuned tides or due to poor non-tidal ssh from the forcing model.

pg 9, lines 10-11: How was 30 minutes of phase lag measured? Is this associated with a particular tidal constituent?

pg 9, lines 13-23 and Figures 5,6: Terminology switches between flow speeds and tidal flow. Suggest to be precise here, tidal currents are typically extracted via tidal analysis and there is no mention of such analysis.

pg 17: How was the residual circulation calculated? Through an average or through a detiding procedure?

Figure 15: vectors are very dense, consider replotting with fewer vectors so the current field is more visible

Figure 15: there is a gap in vectors along the Daliao River; is that a plotting artifact?

pg 19, line 14: "The ecological degradation of wetlands in the LRE has become more and more severe in the past decade." This line difficult to reconcile with the first line of the abstract ("The wetland of Liao River Estuary in northeast China is one of the

best-preserved wetlands across the globe")

pg 19, line 6,7: The salinity appears to vary considerably between dry and wet season at PBW, is a 4 PSU increase in the wet season enough to affect the species here?

Technical corrections

pg 7, line 11: which DEMs were used?

pg 7, line 16-17: are internal and external switched here?

pg 8: Which simulation was validated? Presumably it was case 1 or 3?

---

## Author Comment (AC1) · 13 Oct 2020

**Response to Referee 1**

Dear Professor Pringle:

Thank you very much for your careful review and valuable comments. We have studied your comments carefully and tried our best to revise the manuscript. The point to point responses to your comments are listed as following:

**Question 1.** The authors are very liberal with their use of acronyms making some parts hard to follow. Only need to use acronym if the word is repeated many times and is long. I recommend to just use an acronym for the Liao River Estuary (LRE) and the vegetation (P.a. and S.h.), and spell everything else out.

**Response 1:** Thank you for your kind suggestion on use of acronyms. We have corrected acronyms in the manuscript and only used an acronym for the Liao River Estuary (LRE) and vegetation types (*Suaeda heteroptera*, *Phragmites australis*).

**Question 2.** Page 3, Line 14: What types of models are "ELCIRC and COAWST"?
**Response 2:** Sorry for the carelessness, these questions have been revised.
Gong (2011, 2018) employed the Environmental Fluid Dynamics Code (EFDC) and the Coupled-Ocean-Atmosphere-Wave-Sediment Transport (COAWST) to investigate the effects of local and remote winds and wind waves on salt intrusion in the Modaomen and Pearl River estuaries, respectively, with conclusions having an applicability to other partially mixed estuaries under the threat of salt intrusion.

**Question** 3. Page 4, Lines 5-8: "Estuarine salinity has a significant effect on the growth of coastal wetland plants plays an important role in maintaining the ecological health of estuarine wetlands. Despite this, studies on the spatial and temporal distribution of salinity in estuarine wetlands are limited, with most work generally focusing on salinity transport and saltwater intrusion mechanisms along the estuaries". These two sentences seem to come out of nowhere and do not have any references to back up the statements. Please expand on these sentences and provide details of the previous studies and their limitations.
**Response 3:** Thank you for your valuable advice. We have expanded the content between those sentences and added some references to back up our statements.
Estuarine salinity has a significant effect on the growth of coastal wetland plants and plays an important role in maintaining the ecological health of estuarine wetlands (Song et al. 2009). Despite this, studies on the spatial and temporal distribution of salinity in estuarine wetlands are limited, with most work generally focusing on salinity transport and saltwater intrusion mechanisms along the estuaries (Ralston et al., 2008; Haralambidou et al., 2010; Veerapaga et al, 2019; Wang et al., 2019).

**Question 4.** Page 4, Lines 11-12: "According to satellite images and on-site surveys, the wetlands in the LRE have experienced severe degradation over the past decade". What surveys and satellite images are these? Are these from the authors' own studies? Provide more details.

**Response 4:** We have added reference including the satellite images of the past decade to show the degradation of wetlands in the LRE.

According to satellite images and on-site observations, the wetlands in the LRE have experienced severe degradation over the past decade, particularly in the Pink Beach Wetland (PBW) close to the northwest of Panjin Port (Wang et al., 2020). We have made a series of field surveys for wetland degradation and carried out studies about remote sensing image interpretation in LRE.

**Question 5:** Page 4, Lines 14-15: "Studies have revealed that salinity increases in water and soil can result in the death of S.h. vegetation." What studies are these? Please provide references. Furthermore, what about P.a. vegetation? No details on P.a. are provided in this paragraph.

**Response 5:** Thanks for reminding. We have added some references of studies about salinity increase in soil and water can result in death of *Suaeda heteroptera* seedlings. The expression of S.h. and P.a. plants has been improved in the manuscript.

Experiments implemented by Li et al (2018) have revealed that high salinity can significantly restrain the growth of salt mash vegetations (*Suaeda heteroptera* and *Phragmites australis*).

**Question 6:** Page 5, Line 4: "(2) explore the internal mechanisms of these effects". What does internal mechanisms mean here exactly?

**Response 6:** The original statement in the paragraph was inaccurate. We want to explore the potential influence of these variations (runoff decrease resulting from river closure projects and shoreline changes resulting from the port construction) on growth of *Suaeda heteroptera* in tidal wetlands in the LRE. We have corrected the description in the manuscript.

(2) explore the potential influence of these variations on growth of *Suaeda heteroptera* in tidal wetlands in the LRE;

**Question 7:** Page 6, Lines 16-17: "The application of terrain-following coordinates results in an improved capacity to solve complex bathymetric conditions compared to other existing models." This is not a correct statement. Many other models use terrain-following coordinates (e.g., ADCIRC, SELFE/SCHISM, ROMS) and they have a well-known issue associated the computation of the pressure gradient term in high gradient regions (Haney, 1990) that other researchers have attempted to alleviate (e.g., SELFE/SCHISM uses hybrid coordinates (Zhang et al., 2015).

**Response 7:** Thank you for your correction. It was an improper description, we have corrected it in the manuscript.

**Question 8:** Page 7, Lines 6-10. How did the coastlines from Google Earth differ from the coastlines from Landsat Images? Please explain in more detail about the coastline extraction process (what tools?). Where were the Landsat images used and where were the Google Earth images used?

**Response 8:** The satellite image of LRE region in 1995 showed in Google Earth was blurred, so we chosen the Landsat Image downloaded from Geospatial Data Cloud (http://www.gscloud.cn/sources/?cdataid=263&pdataid=10) to obtain the coastline data of the LRE in 1995. The shoreline of 1995 was manually extracted by adding points on Landsat image

using the ArcGIS software. The remaining coastline of the LRE in 2019 was manually extracted from Google Earth (with higher spatial resolution).

**Question 9:** Page 7, Lines 10-11. "available DEM datasets" does not explain anything. Provide the source.
**Response 9:** The bathymetry of the computation domain was derived from the Navigation Guarantee Department of the Chinese Navy Headquarters (China Navigation Publications Press) and from the topographic survey data in the upstream of LRE by us.

**Question 10:** Page 7, Lines 14-15. Does the spatial resolution model grid vary only with distance between the open boundary and the wetland? Or is there some bathymetry depth function involved as well?
**Response 10:** In our work, the spatial resolution of model grid varies only with distance between the open boundary and the wetland coastline. It is not related to the bathymetry depth, and the mesh number of this area is large, and the precision can meet the calculation requirements.

**Question 11:** Page 7, Lines 16-17. Says the internal mode time step is 2 s and external mode time step is 10 s. I think this should this be reversed. (the external mode is the fast barotropic mode and should have a smaller time step).
**REPONSE 11:** Thank you for your correction. The internal mode time step is 10 s and external mode time step is 2 s.

**Question 12:** Page 7, Line 18. What is the "validated Bohai Sea Parent Model grid"? Any reference? Explain in more detail please.
**Response 12:** The part has been revised in this manuscript. We established a Bohai Sea tidal model (covers the whole LRE region) using the MIKE 21 hydrodynamic model. The tidal model was validated with observed tide level data. Then we calculated the water elevation time series of each open boundary node using the validated Bohai Sea model (unpublished) and applied as open boundary condition in the FVCOM model.

**Question 13:** Page 7, lines 18-19. What do you mean by "salinity was set to 34 PSU at the sea surface and interpolated along the sigma layers"? Interpolated between 34 PSU and what other value?
**Response 13:** Thanks for reminding. The open boundary for salinity was set to 34 PSU at the sea surface and interpolated between 34 and 32 PSU along the sigma layers.

**Question 14:** Page 8, Lines 12-13 & Line 16. Are the open boundary water level conditions inaccurate? Why open boundary conditions may be inaccurate for certain periods? You should be able to quantify this from the "validated Bohai Sea Parent Model grid".
**Response 14:** Thank you for your advice. In fact, we have validated the computational accuracy for the Bohai Sea Large Model by available field data. Due to the large calculation area, there may be some errors in the results of the large model. Generally, the simulation error is acceptable.

**Question 15:** Page 12, Lines 11-12. I don't really see that much evidence in Figures 8 and 9 that

"during the spring tide period, saltwater intrusion distance in the estuary increases compared to that during the neap tide period." Can you be more specific about where you see that? Including a panel showing the differences may help.

**Response 15:** Thanks for your suggestion. We have modified the figures you mentioned for a clearer description. See figs 7 and 8.

**Question 16:** Page 12, Line 24 – Page 13 Lines 1-2. It's unclear to me how the Popescu et al. (2015) reference relates to this sentence. I searched for keywords "salinity" and "salt" in their article and nothing comes up. Furthermore, this sentence implies that increasing salinity definitely inhibits growth of S.h. and P.a. vegetation, but the introduction on Page 4 lines 10-11 says that the "S.h and P.a. are the most common pioneer salt-tolerant plant in the LRE wetlands". Of course this does not mean that a very high level of salinity can't inhibit their growth, but this ties back to my earlier specific comment 5) where I think you need to be more clear and careful about the statements relating to how both S.h. and P.a. are thought to be influenced by salinity, and make the correct citations.

**Response 16:** Sorry for the carelessness. We made a mistake that the Popescu et al., (2015) reference was unrelated to the sentence. We have corrected our statement and made correct citations.

Furthermore, the main factor limiting the growth of *S. heteroptera* is water salinity, and the most suitable salinity for its growth is about 15 psu. If salinity is lower or higher than 15 psu, its growth will be degraded or inhibited. We think that the change of salinity caused by port construction has a potential effect lead to the degradation of estuarine wetland communities.

**Question 17:** Most of the beginning of Section 4.2 (page 14) should be in the introduction and methods section. The first ~5 lines of Section 4.1 and 4.3 are the same. Please only focus on including results in Section 4.

**Response 17:** Thank you for your advice, we have adjusted the structure of Section 4.

**Question 18:** Page 15 Line 7: "This indicates the intensification of the shoreline change with the intrusion of salt water". This needs to be reworded, do you mean that the shoreline change increases the intrusion of salt water onto the wetlands?

**REPONSE 18:** We have reworded the sentence that the shoreline change increases the intrusion of salt water onto the wetlands.

This indicates the intensification of the shoreline change with the intrusion of salt water.

**Question 19:** Page 17, Line 10: says both tidal flow and residual flow were analyzed but only results for the residual flow are presented.

**Response 19:** Sorry for the negligence. We have revised flow analysis in the manuscript, we mainly focus on tidal flow analysis has been deleted.

**Question 20:** It would be nice to use colormaps that are more physically intuitive and unbiased instead of the rainbow ones adopted; refer to Thyng et al. (2016) for colors design to be used for salinity and depths etc.

**Response 20:** Thanks for suggestion on using colormaps. We have redrawn the figures in the

manuscript and changed their colormaps.

Technical Corrections:

**Question 1:** Page 15 Line 5: Figure 12 should be Figure 13.

**Response 1**: this problem has been revised in this manuscript.

**Question 2**: Page 15, Line 14: Fig. 13 should be Fig. 14.

**Response 1**: this problem has been revised in this manuscript.

References

Haney, R.L., 1990. On the Pressure Gradient Force over Steep Topography in Sigma Coordinate Ocean Models. J. Phys. Oceanogr. 21, 610–619.

Thyng, K.M., Greene, C.A., Hetland, R.D., Zimmerle, H.M., DiMarco, S.F., 2016. True colors of oceanography: Guidelines for effective and accurate colormap selection. Oceanography 29, 9–13. doi:10.5670/oceanog.2016.66

**Response**: The reference has been added in this manuscript.

Haney, R.L., 1990. On the Pressure Gradient Force over Steep Topography in Sigma Coordinate Ocean Models. J. Phys. Oceanogr. 21, 610–619.

Thank you for your evaluation of the manuscript.

The authors would like to revise this manuscript if reviewers have any other questions.

Sincerely

authors

---

## Author Comment (AC3) · 13 Oct 2020

**Response to Referee #3**

Dear Reviewer:

Thank you very much for your careful review and valuable comments. We have studied your comments carefully and tried our best to revise the manuscript. The point to point responses to your comments are listed as following:

**Introduction:**

**Question 1:** The model setup section is in need of an overhaul to better describe what has been done here (details below), and more broadly the manuscript should receive some reorganization to get all of the right content into the right sections. For example the start of section 4.1 has introduction material (pg 12, lines 1-5), and section 4.2 also has introduction material (pg 14, lines 8-18), while at pg 14, lines 18-21 look like they belong in a methods section..

**Response 1:** Thank you for your kind suggestion, we have revised the related questions in the manuscript.

**Question 2:** Despite the organizational issues, the manuscript starts off with a focused goal of investigating salinity distribution sensitivity and demonstrates/quantifies the effects of adjusting the domain geometry. In that regard it is successful. Surface forcing is absent so there is a lack of completeless for the model, but this is noted for future work, and presumably explains why specific forecasts about the fate of the salt-tolerant plants is not made.

**Response 2:** Thanks to the reviewers for their valuable comments. The influences of wind stress on the spatial-temporal distribution of salinity in the Liao River Estuary will be the focus of future work. In fact, there are many factors affecting the degradation of *Suaeda Heteroptera* in the wetlands of the Liao River Estuary, such as runoff, rainfall and water pollution. In this paper, we show that the construction of the port will have an potential impact on its salinity environment (*Suaeda Heteroptera*), and it is difficult for us to forecast clearly the fate of the *Suaeda Heteroptera* plant in the tidal wetlands of the Liao River Estuary.

**Specific comments**

**Question 1:** pg 7, lines 17-18: What is the "validated Bohai Sea Parent Model"? Definitely we need some sort of citation here, and some rationale for using it's water level as OBC in the present study. Does this model provide tides only or does it also include non-tidal sea-level variability?

**Response 1:** Thank you for your advice, we have added more accurate description about the model and open boundary conditions in this manuscript. The model was driven by water level derived from the Bohai Sea tidal model using the MIKE 21 hydrodynamic model Model at the open boundary. The Bohai model has been validated using available data as well.

**Question 2: p**g 7, lines 18-19: "The open boundary for salinity was set to 34 PSU at the sea surface and interpolated along the sigma layers." I don't follow; more than one number is needed to conduct an interpolation. Does this mean the salinity OBC was set to 34 PSU at all depths along the entire open boundary? Is the 34 PSU isohaline known to coincide with the model open

boundary? How does that reconcile with estuarine flow characterised by salty inflow at depth and fresher outflow in upper layers?

**Response 2:** Thank you for your questions, we have revised our statement in the manuscript. The open boundary condition for salinity was set to 34 PSU at the sea surface and interpolated between 34 and 32 PSU along the sigma layers, the value was based on data from an unpublished document.

**Question 3:** pg 7, lines 19-20: "Temperature was set to 15◦ C across the whole domain." This sounds like the temperature initial condition. What was done for the temperature OBC? Was temperature an active or inactive tracer in the model? Is water temp mostly uniform in the area justifying setting it to inactive in the model?

**Response 3:** Considering that the average water depth of the LRE is relatively small, variation of temperature in horizontal and vertical directions is negligible, the open boundary condition for temperature was set to uniform 15 ℃ from the surface to the bottom and the initial temperature field was set to a uniform value 15 ℃ across the whole domain accordingly.

**Question 4:** pg 7, lines 20-21: "The initial condition for salinity was based on the steady results derived by running the model for approximately four months." This reads as if the final conditions were used to initialize the model! Presumably the model was started with a different set of conditions (perhaps uniform salinity?) and then the four month run used (perhaps with average runoff?) to produce quasi steady salinity field for initializing the four cases. Please clarify this.

**Response 4:** The initial condition for salinity was based on the quasi-steady results derived by running the model with initially uniform salinity and average runoff for approximately four months.

**Question 5:** pg 7: No information provided about velocity boundary conditions.

**Response 5:** Initial condition: velocity is zero everyhere. In order to calculate the flux at the open boundary, the ghost cells are added at the open boundary in which the velocity is specified as the same value and direction in the open boundary cell.

**Question 6: pg 7:** No justification for not including surface forcing.

**Response 6:** Surface forcing are not considered in this study, we have added this statement in the manuscript.

**Question 7:** pg 7, lines 21-23: How were these numbers selected? Are there discharge records that were used? Pg 6 line 3 mentions low average and high averages of 101 & 285 m³/s for LR but a value of 25 is used for cases 1 & 2. Some rationale for the choices would help here.

**Response 7:** Thank you for your suggestion, we have added supplementary specification in the manuscript.

**Question 8:** pg 8, lines 12-15: It is typical to evaluate water level by splitting into tidal and residual/sub-tidal components, this would help in understanding if mismatches are due to poorly tuned tides or due to poor non-tidal ssh from the forcing model.

**Response 8:** that is a good idea, in this paper, the simulation accuracy of tidal level is acceptable

for us. Thank you for your advice, we will carry out the related assessment work in future work.

**Question 9:** pg 9, lines 10-11: How was 30 minutes of phase lag measured? Is this associated with a particular tidal constituent?

Response 9: The description about phase lag in the manuscript was not rigorous in the entire simulated period, but in individual moments.
.

**Question 10:** pg 9, lines 13-23 and Figures 5, 6: Terminology switches between flow speeds and tidal flow. Suggest to be precise here, tidal currents are typically extracted via tidal analysis and there is no mention of such analysis.

**Response 10:** We have revised this statement in the manuscript, tidal analysis has been removed in this study, we mainly focus on residual flow analysis.

**Question 11:** pg 17: How was the residual circulation calculated? Through an average or through a detiding procedure?

**Response 11:** We obtained the residual velocities by an average calculation.

**Question 12:** Figure 15: vectors are very dense, consider replotting with fewer vectors so the current field is more visible.

**Response 12:** We have redrawn this figure in this manuscript.

**Question 13:** Figure 15: there is a gap in vectors along the Daliao River; is that a plotting artifact?

**Response 13:** We utilized the Tecplot software to do the post-prosessing work, the vector plot was not obtained by regular grid interpolation, but was drawn directly by data on unstructured grid center for sparse processing. A gap appeared because of the coarse resolution of the Daliao River grid.

**Question 14:** pg 19, line 14: "The ecological degradation of wetlands in the LRE has become more and more severe in the past decade." This line difficult to reconcile with the first line of the abstract ("The wetland of Liao River Estuary in northeast China is one of the best-preserved wetlands across the globe")

**Response 14:** The total area of the Liao River Estuary wetland is about 1200 km$^2$, part of the wetland in Liao River Estuary has undergone serious degradation. We have revised description in the manuscript.

**Question 15:** pg 19, line 6,7: The salinity appears to vary considerably between dry and wet season at PBW, is a 4 PSU increase in the wet season enough to affect the species here?

**Response 15:** The main factor limiting the growth of *S. heteroptera* is water salinity, and the most suitable salinity for its growth is about 15 psu. If salinity is lower or higher than 15 psu, its growth will be degraded or inhibited. We think that the change of salinity caused by port construction has a potential effect lead to the degradation of estuarine wetland communities.

**Technical corrections**

**Question 1:** pg 7, line 11: which DEMs were used?
**Response 1:** We have revised this part in this manuscript.

**Question 2:** pg 7, line 16-17: are internal and external switched here?
**Response 2:** We have revised this part in this manuscript.

**Question 3:** pg 8: Which simulation was validated? Presumably it was case 1 or 3?
**Response 3:** To validate the model, we adopted grid 2019 with river runoff to perform the simulation of the tidal dynamics and salt transport in LRE in 2018.

Thank you for your evaluation of the manuscript.

The authors would like to revise this manuscript if reviewers have any other questions.
Sincerely
authors

---

## Author Comment (AC2)

**Response to Referee #2**

**Dear Reviewer:**

Thank you very much for your careful review and valuable comments. We have studied your comments carefully and tried our best to revise the manuscript. The point to point responses to your comments are listed as following:

**Introduction:**

**Question 1:** Information about the main circulation features and tidal dynamics of the coastal Liaodong Bay from the literature would be nice (for example the tide is semi-diurnal close to the coast according to Hao et al., 2005).

**Response 1:** Thank you for your kind suggestion on citation about the main circulation features and tidal dynamics of the coastal Liaodong Bay, we have added the related reference in the manuscript.

**Question 2:** But my main concern is that there is nearly no mention to the paper by Qiao et al. (2018), entitled "Numerical study of hydrodynamic and salinity transport process in Pink Beach wetlands of Liao River Estuary, China", which shares co-authors with this paper. In their paper, Qiao et al. apply the Mike unstructured mesh model to the Liao River Estuary. They focus on the hydrodynamic characteristics and salinity transport processes in Pink Beach wetland of the Liao River Estuary, considering the effect of wetland plant on tidal flow. In the present paper, the authors should emphasize what is new in their study (scenarios with and without the port). Why do they use FVCOM instead of MIKE, is there a reason? In their conclusion, the authors of the present paper mention (Page 19) that runoff increases can decrease salinity in estuary waters due to the dilution of freshwater. Is this result really new? In Qiao et al., Figure 20 shows contour maps of salinity in the LRE under different runoffs during the period of highest saltwater intrusion, and the authors conclude that "the larger the river discharge, the stronger the runoff diluting effect", and "when the river discharge is low, less freshwater is mixed into the system and salinity is higher". Page 4 Line 20 : Sources of data and where they can be downloaded should not be in the introduction but in the Method part.

**Response 2:** Thanks for reminding. In this paper, we want to explore the potential influence of anthropogenic activities (runoff decrease resulting from river closure projects and shoreline changes resulting from the port construction) on degradation of *Suaeda heteroptera* in tidal wetlands in the LRE. The work by Qiao et al (2018) is of great reference value to our work, the innovation in our study is scenarios simulation with and without the port. We applied FVCOM instead MIKE 21 model because we want to investigate the three-dimensional distribution of salinity in the LRE. In mike model, Qiao et al. discussed the influence of runoff decrease on salt intrusion, in this study, we analyzed the influence of shoreline changes on salt intrusion in LRE. According reference's suggestion, we have revised the relevant content in the manuscript.

**Model description:**

Question 1: It would be nice to have more details about the model equations and how they are

solved. At least mention that it is a 3D primitive equation model.

**Response 1:** Thank you for your advice. Due to the limitation of the paper, the model equation and the solution method will not be discussed in this manuscript, however we add more introduction about the 3D primitive equation model.

The FVCOM was adopted to simulate tidal flow and salinity in the LRE. It is a three-dimensional ocean model which was originally developed by Chen et al. (2003) and improved by researchers at the University of Massachusetts-Dartmouth (UMASSD) and Woods Hole Oceanographic Institution (WHOI) (Chen et al., 2006).

**Question 2: Page 6 Line 17: please add details about the vertical coordinates (sigma?) and provide reference.**

**2:** Thank you for reminding, we will add some details about the sigma coordinates.

FVCOM is originally coded for sigma-coordinates in the vertical direction (Chen et al., 2006), the application of terrain-following coordinates results in an improved capacity to solve complex bathymetric conditions (Haney, 1990).

**Question 3:** Page 7 Line 1: "other existing models". As many other models use terrain following (sigma) coordinates, what do you mean? Is it models that use z coordinates? You should be more precise here.

3: Thank you for your correction, we have corrected it in the manuscript.

The application of terrain-following coordinates results in an improved capacity to solve complex bathymetric conditions.

**Model configuration:**

**Question 1:** Model initialization and forcing need to be more detailed. Please give the model initial condition and the boundary conditions for the tides, or add a reference. Also, explain your choices for open boundary salinity of 34 PSU, initial temperature of 15°C, and river discharge scenarios (are values chosen from observation, literature?).

**Response 1:** Thank you for your suggestion. We have added more details about model initialization and forcings. The open boundary condition for salinity was set to 34 PSU at the sea surface which was obtained from an unpublished document and interpolated between 34 and 32 PSU along the sigma layers. Considering that the average water depth of the LRE is relatively small, variation of temperature in horizontal and vertical directions can be ignored, the initial temperature field was set to a uniform value 15  $\mathbb{C}$  across the whole domain accordingly.

**Question 2:** Page 7 Line 11: Please provide a reference for surface water model system. As you do not use the acronym SMS after, do not use it here.

**Response 2:** Thank you for your advice. We have revised this part and added a reference in this manuscript.

**Question 3:** Page 7 Line 18: A reference for the Bohai Sea Parent Model validation would be nice. **Response 3:** As a matter of fact, the Bohai Sea Parent Model mentioned in the paper was an unpublished work of ours. We established a tidal model for the Bohai Sea using the MIKE 21 hydrodynamic model. We will correct our statement about the Bohai Sea Parent Model and

provide more details about open boundary elevation in the manuscript.

**Question 4:** Page 9 Line 4: You could add "by taking into account the bias and correlation between model and observation" to the description of skill sentence. **Response 4:** Thank you for your correction.

**Question 5:** Page 9 Line 11: you claim that "significant errors are observed between the simulated high and low tide levels and observed values". If this is from figure 4, it is not very clear to me. Also, T1 and T2 are very close to the boundary, what is the tidal forcing at the boundary? You explain the poorer fitting results at spring tide by the choice of open boundary conditions, so you may definitely give them in the model description/configuration part. It would be great to have this kind of comparison close to the LR, can we assume that there is no data there?

**Response 5:** The water elevation was chosen as the open boundary condition. What we want to express was that the cause of the simulation error was objectively related to the open boundary condition. We will make a more precise explanation in the manuscript.

Unfortunately, in this study area, only the observed data given in this paper are available during the simulated period, and no other data are available for model validation.

**Results and discussion:**

**Question 1: Page 12 Line 6: Why did you choose 50 hours for averaging?**

**Response 1:** We chosen an average salinity value of 50 hours to reflect the irregular semi-diurnal tidal characteristics of the LRE.

**Question 2: Page 17 Line 13-19: Figure 15 is not clear and should be reworked, as this part is not very clear for the moment.**

**Response 2:** Thank you for your advice. We will redraw Figure 15 and make it clear for reading. See fig.14.

**Question 3: Page 17 Line 13-19: Figure 15 is not clear and should be reworked, as this part is not very clear for the moment.**

**Response 3:** Thank you for your kind suggestion. We will adjust the relevant content in the manuscript. See fig.14.

**Question 4:** Page 18 Line 1-4: As this part deals with the effect of shoreline change on tidal flow, all the text beginning by "In summary" could be moved to the conclusion, or the link with salinity could be added in the subtitle.

Response 4: We mainly focus on residual flow analysis.

**Conclusions:**

**Question 1:** Page 18 Line 10: what is a well-validated model? Maybe you could use a term that refers to the robustness of the model ("proven model"?).

Response 1: We will delete the word 'well-validated'.

**Figures:**

**Figures 1 and 2** Could be merged to give the location of the area at first, and the names of big cities could be added (at least Dalian) to facilitate the reading by foreign scientists.

**Response:** Thank you for your suggestion, we have merged two figures into one, the big cities were added into this manuscript.

See fig. 1.

**Figure 2:** Blue triangles are not visible on the plot. Please add a Table with the coordinates of the stations.

**Response:** Thank you for your suggestion, we have revised this question, please see table 2.

**Figure 4:** It would be nice to have the amplitude and phase for the main tidal components for the comparison in an additional Table or in the text.

**Response:** The harmonic analysis was also performed in this study, the model results are in good agreement with the observations for the main component M2 in the two stations. Where the differences between the model and field data are smaller than 9 percent for the amplitude and 10 degrees for the phase.

**Figure 6:** Why did you choose the scale 18-24 PSU? Is it possible to zoom in? **Response:** this problem has been revised.

**Figures 8 and 9:** Is it necessary to show both surface and bottom maps, as they look very similar? You do not comment the differences in the text so I suggest to remove bottom plots.

**Response:** Thank you for your suggetion, we have removed the bottom plots and replaced by vertical averaged plots.

**Figure 15:** This figure has to be reworked, as it is very hard to see anything, especially the direction of arrows. Perhaps reducing the number of arrows and zooming in areas of interest could help?

**Response:** Thank you very much for your careful advices about modifications of Figures 1, 2, 4, 6, 8, 9 and 15. We truly think your suggestions are very helpful and we will make relevant alterations to these figures.

**Minor comments and Typos:**

**Page 3 Line 14: What are ELCIRC and COAWST? Models?**

**Response:** This problem has been revised. The two models should be the Environmental Fluid Dynamics Code (EFDC) and the Coupled-Ocean-Atmosphere-Wave-Sediment Transport (COAWST).

Page 7 Line 11: Please write "digital elevation model" instead of DEM.

**Response:** this problem has been revised.

Page 7 Line 12: "elements were respectively". **Response:** this problem has been revised.

Page 8 Line 11-12: Delete n comprehensive model validation was performed using the

observation data", as it has already been said at Line 8. Response: this sentence has been deleted.

Page 12 Line 23: Please write "Liaodong Bay" instead of LDB. **Response:** this problem has been revised.

**Page 15 Line 5: I suggest to replace "above" with "upstream". Response:** thank your for your suggestion, we have replaced the word.

Page 15 Line 5: Replace "Figure 12" by "Figure 13". **Response:** this problem has been revised.

Page 17 Line 15: By "its", do you mean "the port"?Response: this problem has been revised.Following the port establishment, ......

Page 17 Line 19: Add "(not shown)" for the comparison of results at neap and spring tides. **Response:** this problem has been revised.

Typos: Page 2 Line 21: "to understand" Page 8 Line 7: "In order to" Page 9 Line 14: "Figure 5 and Figure 6" Page 12 Line 7: "Figures" Page 12 Line 9: "decreases" Response: thank you for your suggestion, this problems have been revised.

We sincerely appreciate your kind suggestions and corrections on our manuscript. The authors would like to revise this manuscript if reviewers have any other questions. Sincerely authors

---

## Author Response (AR2)

**Response to Referee 1**

Dear Professor Pringle:

Thank you very much for your careful review and valuable comments. We have studied your comments carefully and tried our best to revise the manuscript. The point to point responses to your comments are listed as following:

**Question 1.**

Page 8, Line 18-20: The modified sentence is still not really correct. Please change to something like: "FVCOM uses terrain-following vertical sigma coordinates that has the capacity to solve the flow over complex bathymetric conditions (Chen et al., 2006)." (Do not need the Haney (1990) reference)

**Response 1:** Thank you for your kind suggestion. We have corrected the cited reference in the manuscript.

The application of terrain-following coordinates results in an improved capacity to solve complex bathymetric conditions (Chen et al., 2006b).

Thank you for your evaluation of the manuscript.

The authors would like to revise this manuscript if reviewers have any other questions.

Sincerely

authors

**Response to Referee #2**

Dear Reviewer:

Thank you very much for your careful review and valuable comments. We have studied your comments carefully and tried our best to revise the manuscript. The point to point responses to your comments are listed as following:

**Introduction:**
1/ On the salinity boundary condition: 34 PSU at surface at 32 PSU at depth? I would have expected the opposite -- estuarine flow is usually characterised by salty flow at depth and fresher flow at the surface -- so 32 at surface at 34 at depth. Given the constant temperature, does the higher-salinity above lower-salinity water lead to unstable stratification?

**Response 1:** Thank you for your correction, we made a mistake in proofreading the manuscript. The open boundary condition for salinity should be 32 PSU at the sea surface and interpolated between 32 and 34 PSU from the surface to the bottom. The water depth in the study area is shallow, and there is no obvious stratification of water temperature, so we give a constant value in these simulations.

2/ On the river discharges. The authors present two discharge scenarios (high flow season and low flow season) for both pre-port and post-port domains. To me, the 1995 version represents a baseline salinity regime that the vegetation is compatible with -- it survives both seasons, perhaps thriving in one and tolerating the other. The suite of experiments does not take into account the decrease in freshwater discharge due to recent diversions for use by the population. That is, there should be a low flow 1995 scenario contrasted with a low-flow 2019 scenario where the low-flows are adjusted for the diversions. The manuscript would benefit from drawing attention to the fact that this comparison was not done (perhaps saving it for future work to be done alongside adding surface forcing).

**Response:** Thank you for your kind suggestion. We have made a comparison of salinity calculated in low flow 1995 scenario and low flow 2019 scenario in Figure 12 (a) and (c), in Section 4.2.

Meanwhile, the speculation in section 4.1 about higher salinity as a consequence of increased water use seems to have been promoted to become part of the conclusions in section 5 ("Once the port was constructed, its obstruction of the port area strengthened the tidal current mixing, and the partial runoff from the Daliao River was diverted. As a result, the fresh water dilution effect further weakened, thus increasing the salinity of the sea water in the lower reaches of the Pink Beach Wetland."). The latter part wasn't shown, so the conclusions needs to be adjusted here.

**Response:** We have added description about the conclusion from Section 5 in Section 4.3, as a supplementary discussion about salinity increase in Section 4.2. Conclusions in Section 5 have also been adjusted accordingly.

2b/ Also regarding Table 1: are the high/low flow season discharges from 1995 (pre-port and pre-diversions) or from 2019 (post-port and post-diversions)?

**Response 2b:** We adopted a multi-year mean river discharge as a normal mean value (the high one), then we hypothetically set the low discharge values of both rivers to reflect a sharply reduced river discharge scenario.

We sincerely appreciate your kind suggestions and corrections on our manuscript. The authors would like to revise this manuscript if reviewers have any other questions.
Sincerely
authors

**Response to Referee #3**

Dear Reviewer:

Thank you very much for your careful review and valuable comments. We have studied your comments carefully and tried our best to revise the manuscript. The point to point responses to your comments are listed as following:

Introduction

Question 2

The authors answered "We applied FVCOM instead MIKE 21 model because we want to investigate the three-dimensional distribution of salinity in the LRE." I do not understand: do thay mean that MIKE21 does not enable 3D studies? The authors add "In mike model, Qiao et al. discussed the influence of runoff decrease on salt intrusion", so I am a little confused.

**Response:** Sorry for the inappropriate reply. The MIKE 21 model utilizes two-dimensional shallow water equations and simulates water level variation and flow with depth-averaged unsteady two-dimensional free-surface flows within the study area. FVCOM module is an unstructured grid, finite-volume, three-dimensional, primitive equations coastal ocean model, which was originally developed by Chen et al. (2003). It has been widely applied for researches in large lakes, estuarine regions, and regional ocean areas. In this study, we use this 3D model to investigate the three-dimensional distribution of salinity in the LRE.

Model description

Question 1

I am still frustrated with the model description, and I do not find any mention of the use of primitive equations. I suggest the author consider the following description:

Page 8 Line 12.

"It is a three-dimensional ocean model (Chen et al. 2003, http://fvcom.smast.umassd.edu/fvcom) based on primitive equation and using the finite volume method with the capability to deal with horizontal unstructured triangular cells, which is well fitted to irregular coastline."

**Response:** Thank you for your suggestion, we have modified the model description using your sentence.

Model configuration

Question 1

The authors partially answered the question. They tackled the issue of initial and boundary variables values but they did not add the description of open boundary conditions. FVCOM offers the choice between different OBC schemes. For the elevation, did the authors use Orlansky conditions? Radiative GWI? What about the heat and momentum OBC?

**Response:** Thank you for your reminding. We have added OBC schemes in the Model configuration: The open boundary condition for surface elevation was original FVCOM setup (ASL) and that for the perturbation of salinity and temperature was Blumberg and Khanta (BKI) condition (Blumberg and Kantha, 1985). No surface heat flux was considered in this paper.

Question 5

Your answer mentions some additional explanations. I was not able to find them in the text, nothing was added in that paragraph. Please help the reviewer in finding them.

**Response**: Sorry for the negligence in proofreading the manuscript. We have added additional explanation: the velocity in the open cell is computed using the linear or nonlinear momentum equations.

Results and discussion

Question2

Figure 14 is a zoom of Figure 15, it represents a great improvement but it is still not easy to handle. You should definitely put less arrows but enlarge them.

**Response:** We follow your suggestion and made some little adjustments, Figure 14 have been redrawn.

Question 4

If your focus is mainly on residual flow, then you should not put a summary on salinity results.

**Response:** Thank you for your suggestion. We have adjusted description in Section 4.3, trying to illustrate the reason for salinity increase in downstream of Pink Beach Wetland via residual flow analysis.

Figures

Figure 6 looks the same than before

**Response:** We have redrawn Figure 6.

Minor corrections and typos:

Page 8 Line 4: Split this long sentence into two sentences, replace with "semidiurnal tide. In addition,…".

Page 8 Line 19. Split into two sentences, replace with "(Chen et al., 2006). The application…"

Page 9 Line 14: replace with "of the LRE that we measured".

Page 9 Line 22: remove "Model", and replace "zero velocity" with "ocean at rest, zero velocity".

Page 20 Line 17: the doi link does not work for Chen et al. (2003), replace with a valid one.

Page 12 Line 6: there is a problem with the sentence beginning with "Where". Please correct.

Page 19 Line 11: "differs"

**Response:** Sincerely thank you for your corrections. We have revised sentences and typos follow your suggetions.

Thank you for your evaluation of the manuscript.

The authors would like to revise this manuscript if reviewers have any other questions.

Sincerely

authors